# Overt Word Reading and Visual Object Naming in Adults with Dyslexia: Electroencephalography Study in Transparent Orthography

**DOI:** 10.3390/bioengineering11050459

**Published:** 2024-05-04

**Authors:** Maja Perkušić Čović, Igor Vujović, Joško Šoda, Marijan Palmović, Maja Rogić Vidaković

**Affiliations:** 1Polyclinic for Rehabilitation of People with Developmental Disorders, 21000 Split, Croatia; maja.perkusic.covic@mefst.hr; 2Signal Processing, Analysis, and Advanced Diagnostics Research and Education Laboratory (SPAADREL), Faculty of Maritime Studies, University of Split, 21000 Split, Croatia; ivujovic@pfst.hr (I.V.); jsoda@pfst.hr (J.Š.); 3Laboratory for Psycholinguistic Research, Department of Speech and Language Pathology, University of Zagreb, 10000 Zagreb, Croatia; marijan.palmovic@erf.unizg.hr; 4Laboratory for Human and Experimental Neurophysiology, Department of Neuroscience, School of Medicine, University of Split, 21000 Split, Croatia

**Keywords:** dyslexia, developmental dyslexia, evoked response potentials (ERP), reading aloud, naming aloud, overt reading, overt naming

## Abstract

The study aimed to investigate overt reading and naming processes in adult people with dyslexia (PDs) in shallow (transparent) language orthography. The results of adult PDs are compared with adult healthy controls HCs. Comparisons are made in three phases: pre-lexical (150–260 ms), lexical (280–700 ms), and post-lexical stage of processing (750–1000 ms) time window. Twelve PDs and HCs performed overt reading and naming tasks under EEG recording. The word reading and naming task consisted of sparse neighborhoods with closed phonemic onset (words/objects sharing the same onset). For the analysis of the mean ERP amplitude for pre-lexical, lexical, and post-lexical time window, a mixed design ANOVA was performed with the right (F4, FC2, FC6, C4, T8, CP2, CP6, P4) and left (F3, FC5, FC1, T7, C3, CP5, CP1, P7, P3) electrode sites, within-subject factors and group (PD vs. HC) as between-subject factor. Behavioral response latency results revealed significantly prolonged reading latency between HCs and PDs, while no difference was detected in naming response latency. ERP differences were found between PDs and HCs in the right hemisphere’s pre-lexical time window (160–200 ms) for word reading aloud. For visual object naming aloud, ERP differences were found between PDs and HCs in the right hemisphere’s post-lexical time window (900–1000 ms). The present study demonstrated different distributions of the electric field at the scalp in specific time windows between two groups in the right hemisphere in both word reading and visual object naming aloud, suggesting alternative processing strategies in adult PDs. These results indirectly support the view that adult PDs in shallow language orthography probably rely on the grapho-phonological route during overt word reading and have difficulties with phoneme and word retrieval during overt visual object naming in adulthood.

## 1. Introduction

Adult dyslexia affects about 4% of the population, while in childhood, it occurs in 5–17% of the pediatric population [1,2]. Dyslexia is defined as persistent reading difficulty affecting male and female subjects equally [2], and sensory deficits, cognitive deficits, lack of motivation, or lack of adequate reading instruction cannot explain its symptomatology. According to the International Dyslexia Association, dyslexia is defined as a specific learning disability with a neurobiological origin and is characterized by difficulties with accurate and/or fluent word recognition and poor spelling and decoding abilities [3]. These difficulties typically result from a deficit in the phonological component of language that is often unexpected in relation to other cognitive abilities [1]. Deficits at the level of the phonologic module impair the ability to segment the written words into their underlying phonologic elements [1]. The Diagnostic and Statistical Manual of Mental Disorders, Fifth Edition (DSM-5) classifies dyslexia as a form of neurodevelopmental disorder and defines dyslexia as a difficulty in learning to decode (read aloud) and to spell [4]. Dyslexia can occur across the IQ range, and poor decoding skills require the same kind of intervention regardless of IQ [4]. Therefore, according to DSM-5, the term “specific learning disability (difficulty)” has been replaced with “difficulty in learning to decode”. Secondary consequences may lead to problems in reading comprehension and reduced reading experience that can impede the growth of vocabulary and background knowledge [3]. The diagnostics of dyslexia are troublesome when comparing different countries with respect to the tools used in the definition and support for people with dyslexia (PD) at local and regional sites [5,6,7,8,9,10].

The fundamental prerequisite skills necessary for reading ability are phonological processing skills (phonological awareness and phonemic awareness) that begin to develop even before the beginning of formal schooling and continue through third grade and beyond. Phonological processing is usually described as having three main components that can be detected in PDs as: poor phonological awareness (the ability to access and manipulate speech sounds consciously), slow lexical retrieval evidenced in rapid automatized naming tasks [11,12], and poor verbal short-memory [13,14]. Some authors found that deficits in lexical retrieval [12,15], verbal short-term memory, and naming processing speed persist into adulthood [16,17]. Phonological deficits can be regarded as a valid explanation for dyslexia symptoms in a wide variety of spoken languages and writing systems [18], however, it is still a great challenge to differentiate underlying processes involved in word recognition and reading [19,20,21,22,23]. Reading can be explained as a complex cognitive process of decoding orthographic symbols (orthography) into corresponding graphemes/phonemes (phonology) of language to link these symbols with representations of meaning stored in mental lexicon (semantics) [24]. The final goal of the reading process is to comprehend decoded text [25]. Several reading computational models contributed to the understanding of the reading process: The dual route cascaded model of visual word recognition and reading aloud (DRC) [19], the triangle model [26], and the connectionist dual-process model (CDP) [27]. According to a mostly accepted DRC model [19], reading can be achieved through two distinct routes: the direct semantic route (ventral route) and the indirect grapho-phonological route (dorsal route). Furthermore, different language processing stages might impact reading and naming processes. Visual and semantic processing is required to accomplish the naming process while reading could be established directly through orthographic and phonological representations, particularly seen in shallow (transparent orthographies), a type of orthography where a word’s written and spoken forms are very similar [19,21]. Different underlying processes are involved in picture naming speed (e.g., visual complexity, frequency, name agreement, age of acquisition, etc.) [28]. Both frequency and age of acquisition influence picture naming speed [28,29]. Further, differences in language processing (especially evidenced in naming tasks) are strongly correlated with language transparency and orthography [30,31,32,33,34]. It is assumed that results on rapid automatized naming tasks can predict reading development in languages with shallow (transparent) orthographies [31,33], while phoneme awareness was found to be the strongest predictor of decoding skills in different languages [30]. 

Regarding findings of electrophysiological and neuroimaging studies, the most frequently reported are structural and functional impairments in dyslexia [34,35,36,37,38,39,40,41,42,43,44,45,46,47]. Aberrant activation reported for PD relates to the left hemisphere brain areas, including the cerebellar area, inferior frontal area, situated near Broca’s area, implicated in motor speech production, a peri-sylvian temporo-parietal region implicated in speech comprehension, and inferior temporo-occipital region (very often referred as the visual word form area) implicated in fast word decoding. Structural impairments in PD are associated with reduced grey matter volume in the left temporo-parietal cortex, decreased white matter connectivity between reading networks, and hypo-activation of the left temporo-occipital cortex and temporo-parietal cortex. Electroencephalographic (EEG) studies have mostly highlighted the role of lower frequency bands in dyslexia, especially an increase in delta and theta waves and a reduction in alpha and beta waves [48]. Homologous structures of abnormally activated hemisphere cortical regions are shown to be hyperactivated or under-activated in the right hemisphere [35,49,50,51,52]. Furthermore, studies employing event-related potentials (ERP) have provided the temporal course of activation of anatomical structures during specific reading tasks (overt and covert reading and naming tasks) [53,54,55,56,57]. In both reading and naming tasks, the analysis of visual characteristics of words/pictures most likely occurs between 50 and 100 ms after stimulus presentation, while the perception of the word/object shape begins at approximately 150 ms and is regarded as a pre-lexical stage [53,58,59,60,61,62,63,64,65,66]. Further, frontal areas become active at approximately 180 ms, and semantic analysis (lexical stage) of words/objects begins at about 200 ms and unfolds around 500–600 ms involving the frontal and left superior temporal cortex and this phase is regarded as a lexical stage [57,60,63,65,67,68,69,70,71,72,73,74,75,76,77,78,79]. The post-lexical stage occurs approximately after the first 500 ms and unfolds around 1000 ms [53,80,81,82,83].

The earliest signs of differences between PDs and healthy control subjects (HCs) during word reading were seen in the pre-lexical stage, approximately around 150 ms to 250 ms post-stimulus onset [21,84], usually indexed by N170 ERP component [21,40,56,84,85,86,87,88,89], in lexical stage around 250 to 600 ms and indexed by N320 [40,56] component and N400 [64,90,91,92] activated by repetition tasks or grapheme to phoneme conversion mechanisms. Differences in the post-lexical stage between PDs and HCs were seen approximately after the 500 ms to 1000 ms post-stimulus onset indexed by P600 ERP component [53,91,93,94] or late positive complex (LPC) [91,95] engaged during the repetition of words/pseudo words and articulatory processes. 

In pre-lexical naming processes, PDs and HCs differ approximately between 170–200 ms to 400 ms post-stimulus (visual object) onset, indexed by N170 and N400 ERP components [54,63,96,97]. In the lexical stage of the naming process, the differences between PDs and HCs were seen around 250 ms to 450 ms indexed by N2, N3, and N400 ERP components [59,61,71,72,98,99,100] post-stimulus onset usually engaged in semantics or phonological processing. In the post-lexical stage of the naming process, the differences between PDs and HCs arise approximately after the first 500 ms [63,83,101,102,103] with subjects engaged in high level visual, conceptual processing.

The majority of ERP studies have been conducted on non-transparent orthographies (i.e., English, French, Portuguese) using covert reading and naming designs in adult PDs [35,40,64,81,87,89] and children with dyslexia [48,104] investigating pre-lexical, lexical and post-lexical stage of processing [35,40,47,59,64,81,87,89,91]. Lately, two studies on dyslexia have been conducted by using an overt design in adult PDs [56] and children with dyslexia [54] in non-transparent orthographies. Mahe et al. [56] investigated electrophysiological correlates of dyslexia in adult French PDs and expert readers engaged in reading aloud and lexical decision tasks. The study reported a similar electrophysiological pattern between groups in lexical decision tasks, while reduced ERP amplitudes in the left hemisphere were found approximately after the first 100 to 300 ms in adult PDs during reading aloud compared to the control group. In line with the study results, it was proposed that the overt reading task might be more suitable to tap underlying phonological deficits of adult PDs, while the lexical decision task would be more predictable to tap lexical access and word retrieval. Furthermore, Bakos et al. [54] investigated the underlying architecture of naming speed differences in 10-year-old children with reading or spelling disorders and typically developing children using digit naming tasks. The study reported reduced ERP amplitudes in the pre-lexical period of approximately 300 ms post-stimulus in the left hemisphere in the reading-disabled group but not in children with isolated spelling disorders.

Overall, research on reading and naming processes in PDs in languages with shallow (transparent) orthographies is still minor [84,105] with ongoing debate which underlying neurocognitive mechanisms are associated with phonological processing and how orthography can affect those processes [18]. The Croatian language is considered to be a language with shallow (transparent) orthography. But, beyond that, in everyday clinical practice, we see students who cannot easily overcome phonological difficulties and have reading and naming fluency problems that persist into adulthood. In line with that, we aimed to capture subtle differences in electrophysiological level in pre-lexical, lexical, and post-lexical processing stages in overt reading and naming in adult PDs compared to control subjects. As far as we know, the present ERP study is the first one exploring the reading and naming process in overt design in shallow orthography in adult PDs.

## 2. Materials and Methods

### 2.1. Participants

Twelve healthy adult control subjects (HCs) (mean age: 29.5 ± 8.9, 6 males, 6 females) and twelve adult participants with dyslexia (PD) (mean age: 22.08 ± 8.4, 6 males, 6 females) participated in the study. Due to technical issues (signal noise, bad signal from two or more electrodes, major motor artifacts), 2 HC subjects were excluded from electrophysiological data analysis. The final sample included 10 healthy HCs (mean age 27.8 ± 8.21, 6 females, 4 males) and 12 adult PDs (mean age 22.08 ± 8.4). No significant difference was found between age (t = −1.03; *p* = 0.31) and gender (χ^2^ = 0.22; *p* = 0.63) between two groups. HC and PD subjects voluntarily participated in the study and were native Croatian speakers. All participants were right-handed dominant, according to the Edinburgh Handedness Inventory [106]. None of the participants had a history of neurological or psychiatric disorders, and they were free of other diagnoses (i.e., attention deficit disorder, dysphasia). PD subjects were diagnosed by a speech and language pathologist (SLP) during their early childhood education. They all had normal or corrected to normal vision. Prior to electrophysiological testing, participants’ cognitive abilities were tested by a cognitive nonverbal test [107], which showed normal findings. Both groups achieved nonverbal normal intelligence results. According to the Cognitive nonverbal test, average normal nonverbal intelligence was in the range −1z-value to 1z-value, 15 centile-85 centile, 40T-value-60T-value. Deviation IQ, according to the Wechsler scale, was 85 to 115. Both groups were within normal IQ findings. 

The study procedure was approved by the Ethical Committee of the Polyclinic for Rehabilitation of People with Developmental Disorders, Split, Croatia, Class: 135-01/17-01/01, No.: 2181-164-17-01.

### 2.2. Materials and Stimuli

The stimuli were 64 words (for the word reading task) and their corresponding color drawings of objects (for visual object naming task) selected from Revisiting Snodgrass and Vanderwart’s pictorial object set [108] and from Corel Gallery^TM^Magic [109]. Picture stimuli are provided in Appendix A. Words and corresponding pictures were taken from the Croatian word frequency dictionary [110] (Appendix A). The length of target words varied from 4 to 8 phonemes and consisted of 2 to 4 syllables. Two reading/naming blocks started with /b/ and /d/ consonants, and two reading/naming blocks with consonants /k/ and /g/. The graphemes /b/, /d/, /g/, and /k/ are chosen because PDs in the Croatian language often make visual or auditive mistakes during reading [111]. Word and picture stimuli were divided into four blocks consisting of 16 words/pictures each (Figure 1). 

### 2.3. Procedure

Participants were tested individually in a quiet room in the EEG laboratory of Polyclinic for People with Disabilities, Split, Croatia. They sat 60 cm in front of the computer screen. Pictures were presented in constant size of 9.5 cm × 9.5 cm on a white screen. Participants were asked to read words overtly and name the pictures overtly presented on the computer screen as fast and accurately as possible. Before the experiment, they were familiarized with the procedure during one practice trial block of words and pictures. The stimuli were presented using Presentation^®^ software (Version 20.0, Neurobehavioral Systems, Inc., Berkeley, CA, USA, www.neurobs.com, accessed on 15 January 2018) [112]. An experimental trial had the following structure: fixation cross was presented for 850 ms, followed by the presentation of picture/word for 2000 ms, and finally, a blank screen was presented for 750 ms on a white background. The participants had 2000 ms to respond verbally in both experimental tasks. Experimental blocks consisted of 16 trials per block (four for word reading and four for visual object naming). There were 64 trials in the word reading task and 64 trials in the visual object naming task for each participant, in sum 128 trials for both conditions.

Figure 1 presents a single trial for word reading and visual object naming tasks. Recording of the subject’s response started at zero when the word/picture was presented to the subject (microphone symbol, Figure 1). The order of picture naming blocks and word reading blocks was randomized across participants. The entire experimental session lasted approximately 45 min for each subject.

### 2.4. ERP Acquisition and Pre-Processing

Continuous electroencephalographic (EEG) activity was recorded with electrodes mounted in an elastic cap (Figure 2) from 22 sites placed according to the International 10/20 system (Natus^®^ EEG32U™, NicoletOneTM EEG system, Planegg, Germany). Participants were asked to avoid eye blinks and to be calm during recording. One additional electrode was placed at the left mastoid for online referencing. A technician directly marked reading and naming epochs on EEG, while the subject’s responses for reading and naming tasks were recorded simultaneously and stored for offline analysis. The recording was controlled using Nicolet EEG study room software (version 5.91.0.248). Two additional electrodes were placed on the participants’ heads (EOG1, EOG2) near the left and right eye for online referencing, and one was placed on the participants’ nose as an off-line reference channel. Furthermore, the EEG was continuously recorded and digitized at 250 Hz, the impedance was kept below 25 kΩ, and the sensitivity was 70 microvolts.

The complete EEG analysis was performed using EEGLAB (version 15.x (dev)) [113] integrated into MATLAB software (9.4.0.949201 (R2018a)) [114]. Before the segmentation, the EEG signal was filtered with a bandpass filter between 1 and 30 Hz. Removing bad channels was performed using the pop_rejchan (EEG) code implemented in MATLAB [114]. In spectral eegplot view, a visual inspection of bad channels proposed to remove was performed. If the proposed channels had an abnormal appearance, abnormal distributions, having major artifacts, noise was removed. After removing bad channels that were identified with spectrum criteria, all data were re-reference again but without rejected channels. Furthermore, epochs of interest were extracted. Bad segments in epochs of interest were removed using code pop_rejmenu (EEG, 1) and an independent component analysis (ICA) algorithm implemented in MATLAB software [114]. From a large variety of ICA algorithms for the purpose of this study, runica (), a function for automated infomax ICA decomposition, was selected [115].

Epochs that consisted of events of interest (related to reading and naming responses) were extracted. Epochs without response (no-response/answer), incorrect response, or having major motor artifacts were excluded from further analysis. The 1999 ms long epochs were averaged in reference to the 1000 ms pre-stimulus baseline correction to ensure that segmented epochs will start with reading and naming epochs and to avoid motor artifacts in epochs without answer and epochs with incorrect answers. Overall ERP analysis consisted of 45–64 epochs (mean 56) in HCs, 45–64 epochs (mean 58) in PDs for word reading task, 40–64 epochs (mean 56) in HCs and 54–64 (mean 59) in PDs for visual object naming task. The number of averaged trials and electrodes did not differ between groups.

The present ERP study explored pre-lexical (150–260 ms), lexical (280–700 ms), and post-lexical stages of processing (750–1000 ms) in both overt reading and naming tasks [40,53,54,56,59,83,102,116,117]. The spatial and temporal topographical analysis of ERPs for reading and naming was conducted using parametric statistical methods incorporated in EEGLAB software. It used multiple channel selection of ERP plotting format in specified time windows. Reading was explored in a pre-lexical stage in the time window 160–260 ms [21,40,56,84,87], a lexical stage in a time window 450–700 ms [40,56,64,90,117], and a post-lexical stage in time window 750–900 ms [53,91,94,116]. Naming processing was explored in pre-lexical stage 150–200 ms [54,63,96,97], lexical stage 280–440 ms [59,71,72,99,100,117] and post-lexical stage 900–1000 ms [63,83,102,103,116]. 

Furthermore, reading and naming ERPs plotted in a scalp array were plotted in EEGLAB software using the analyzed time windows. Before plotting, ERPs were normalized to spectrum criteria using data statistics incorporated in EEGLAB software. Before displaying topographic maps, all channels were checked due to abnormal appearance, abnormal values, abnormal distributions, abnormal trends, and abnormal spectra.

Behavioral responses on reading and naming reaction time (response latency) were estimated in Praat (Version 5.3.56) [118] from the onset of the stimulus (word or visually presented object) to the onset of the subject’s verbal response (Figure 3). Verbal response longer than 2000 ms and wrong answers provided by subjects were excluded from further analysis. 

### 2.5. Statistical Analysis

Skewness and kurtosis parameters were tested for the individually averaged evoked response potential ERP amplitude at each electrode, time point, and reading and naming response latency results. Results indicated acceptable values for the parametric statistic. Descriptive statistics of relevant parameters were summarized by N, mean, and standard deviations. The chi-square test (χ^2^) was used to determine differences in qualitative error frequency between groups. Mean value comparisons were tested using the T-test. For the analysis of the mean ERP amplitude for pre-lexical, lexical, and post-lexical time window, a mixed design Analysis of variance (ANOVA) was performed with Hemisphere (Left vs. Right) (Right: F4; FC2; FC6; C4; T8; CP2; CP6; P4; vs. Left: F3; FC5; FC1; T7; C3; CP5; CP1; P7; P3) as within-subject factors and Group (Control vs. Dyslexia) as between-subject factor. The effect size was presented as Cohen’s d (t-test), eta squared η2 (ANOVA) and Yule’s Q (chi-square test). Fisher LSD post hoc test was further calculated. In all calculations, a *p*-value of <0.05 was considered statistically significant. Data analyses were performed using the software Statistica Soft 12.

## 3. Results

### 3.1. Behavioral Results

Qualitative analysis showed that PDs made significantly more errors in the word reading task (2.47%) compared to HCs (0.52%) (χ^2^ = 9.93, *p* < 0.01, Q = 0.6) and significantly more errors in the visual object naming task (35.54%) compared to HCs (24.60%) (χ^2^ = 21.84, *p* < 0.001, Q = 0.2) (Table 1). In the word reading task, PDs made significantly more phonological errors (1.43%) compared to HCs (0.52%) (χ^2^ = 4.61, *p* = 0.03, Q = 0.5). In the visual object naming task, PDs provided significantly more “no answer” (10.02%) compared to HCs (1.72%) (χ^2^ = 48.34, *p* < 0.001, Q = 0.7) (Table 1). Behavioral analysis of the response latency revealed significant differences in word reading latency (t = 17.81; *p* < 0.001; d = 3.14) between PD and HC subjects, with PDs having prolonged reading latency compared to HCs (t = 0.16; *p* = 0.86). (Table 2).

### 3.2. ERP Results

The topographic view of the electric signal of word reading ERPs (Figure 4) and visual object naming ERPs (Figure 5) are presented in PD and HC subjects’ pre-lexical, lexical, and post-lexical time windows. Descriptive ERP data for amplitude values of the electric signal in the pre-lexical, lexical, and post-lexical window are provided in Appendix A. Topographic plots of ERP data were normalized to mean amplitudes of the electric signal in microvolts in respective time windows.

#### 3.2.1. Reading Task—ERP Results

Notable ERP amplitude differences concerning right vs. left hemisphere for pre-lexical (Fhemisphere = 13.09, *p* < 0.001, *η*^2^ = 0.39; Fgroup x hemisphere = 4.79, *p* = 0.03, *η*^2^ = 0.19), lexical (Fhemisphere = 12.50, *p* < 0.001, *η*^2^ = 0.38), and post-lexical (Fhemisphere = 6.82, *p* = 0.009, *η*^2^ = 0.25) time window were found for word reading (Table 3).

The Fisher LSD *post hoc* test further revealed differences in right hemisphere ERPs between HC and PD subjects for a pre-lexical time window (*p* = 0.02) (Figure 6 and Figure 7) (Table 3). Furthermore, a significant difference was calculated for HC between right and left hemisphere potentials for lexical (*p* = 0.01) and post-lexical (*p* = 0.004) time window and for PD group for pre-lexical (*p* < 0.001) and lexical (*p* = 0.01) time window (Table 3). Grand averaged reading ERPs of ROI electrodes in the pre-lexical stage are provided in Appendix A.

#### 3.2.2. Naming Task—ERP Results

Results for ERPs indicate significant differences between groups and hemispheres for pre-lexical (Fhemisphere = 17.03, *p* < 0.001, *η*^2^ = 0.46), lexical (Fhemisphere = 7.94, *p* = 0.006, *η*^2^ = 0.28), and post-lexical (Fgroup = 6.19, *p* = 0.01, *η*^2^ = 0.23; Fhemisphere = 5.62, *p* = 0.02, *η*^2^ = 0.21) time window for naming task (Table 4). The Fisher LSD post hoc test revealed significant differences for right hemisphere ERPs between HCs and PDs for the post-lexical time window (*p* = 0.02) (Figure 8 and Figure 9) (Table 4). Furthermore, a significant difference was calculated for HC between right and left hemisphere potentials for pre-lexical (*p* = 0.04) and post-lexical (*p* = 0.02) time window and for PDs for pre-lexical (*p* < 0.001) and lexical (*p* = 0.02) time window (Table 4). Grand averaged naming ERPs of ROI electrodes in the post-lexical window are provided in Appendix A.

## 4. Discussion

The main findings of the present study revealed ERP mean amplitude differences between PDs and HCs in the right hemisphere in the pre-lexical time window (160–200 ms) for word reading aloud and in the post-lexical window (900–1000 ms) for visual object naming aloud. 

Behavioral analysis of the word reading latency and naming proved significantly prolonged word reading latency (t = 17.81; *p* < 0.001) between PDs and HCs subjects, which is similar to findings of previously reported studies [36,56,87]. For the naming response latency, no differences were found between PDs and HCs, similar to previously reported findings [34,36]. 

### 4.1. Overt Reading of Words and ERP

Reading aloud requires an explicit grapho-phonological route, also called a non-lexical route [19]. As people progress in reading, they rely more on the semantic route (lexical route) [19]. However, adult PDs probably rely more on grapho-phonological route [15,19,40]. According to cross-cultural studies, reading and naming deficits in developmental dyslexia can be affected by orthographic depth and orthographic consistency [84,119]. Carioti et al. [119] reported that orthographic depth and consistency might have an impact on the manifestations and symptoms of adult PDs in deep (non-transparent) and shallow (transparent) orthographies. Regarding orthographic depth and consistency, the Croatian language is considered a language with shallow (transparent) orthography, such as Italian or Greek. Beyond transparency, the Croatian language has a complex syllabic system, double graphemes, and graphemes with diacritics [111]. 

The present study shows that the word reading mean ERP amplitude for the pre-lexical time window (160–260 ms) was significantly increased over the right hemisphere for adult PDs compared to HCs. Previous ERP studies reported that grapho-phoneme conversion takes place in the pre-lexical window approximately 160–260 ms post-stimuli [40,56,120]. In comparison to the present study results, Mahe et al. [56] found significantly reduced ERP amplitudes in the pre-lexical stage (100 ms post-stimulus) over the left hemisphere in adult PDs compared to HCs during overt word and pseudo words reading in non-transparent French language. Further, previously reported studies on non-transparent languages investigating the pre-lexical stage of processing to orthographic stimuli (reading words) vs. non-orthographic stimuli using covert design (lexical decision judgment task) found greater activation in the right hemisphere for orthographic stimuli in adult PDs [40,87,89,121], as well as in children with dyslexia [54]. The design applied in the present study consisted of sparse neighborhoods with closed phonemic onset (words sharing the same onset, i.e., b–d, g–k) [122,123] that might have had an impact on the prolonged duration of phoneme code retrieval in adult PDs in the pre-lexical stage of processing during overt reading. Further, it is assumed that adult PDs might use an alternative strategy of decoding processes relying on the right hemisphere in the pre-lexical time window [56,121]. Also, according to previously reported findings, adult PDs might have persistent phonological processing difficulties during reading and exhibit a lower score in rapid naming, working memory, and visual-attention [84,124,125,126]. Contrary to PDs, HCs use more efficiently a direct lexical route in decoding familiar words during word reading, while adult PDs rely on a non-lexical route and require extra time for decoding, similar to the decoding process seen in dyslexic children [126] as well in adult PDs [127,128,129]. However, several ERP studies conducted on non-transparent languages using covert design (delayed phonological task, auditory lexical decision task) failed to find differences in the pre-lexical processing stage in adult PDs compared to controls [127,128]. The differences in adult PDs compared to HCs were reported in the post-lexical stage after the first 300 ms in the left hemisphere with reduced ERP amplitudes to words compared to pseudo words during the delayed phonological task and auditory lexical decision task [127,128]. Contrary to these findings, Mahe et al. [56] failed to find differences in the pre-lexical stage of processing during lexical decision task in adult PDs compared to HCs on non-transparent language by using lexical decision task, assuming that overt (reading aloud) task might be more suitable to investigate phonological processing compared to lexical decision task.

To summarize, phonological abilities and reading outcomes are strongly associated in transparent and non-transparent orthographies, but still, there is a gap in cross-cultural studies in investigating systematic differences in the orthographic and phonological characteristics of the languages in dyslexia [18]. According to Zoccolotti [130], future studies could design the type of phonological task associated with specific language orthography to ensure the replicability and comparison of results that would eventually lead to understanding an underlying phonological mechanism in languages with transparent and non-transparent orthographies. 

### 4.2. Overt Visual Object Naming and ERP

Regarding Levelt’s model of visual word production [22], visual object naming tasks can be divided into different stages. During the pre-lexical stage of processing, within first 150 ms, the brain is engaged in visual processing, lexical stage lemma selection (150–275 ms), post-lexical stage phonological encoding (275–400 ms), and articulatory processing (after the first 500 ms). It is believed that reading and naming might share similar neurocognitive processes (word/picture retrieval, storing phonological information, visual processing, semantic analysis) [36,96]. In the reading process, phonological information is retrieved from orthography, compared to visual object naming, where phonological information is retrieved from the semantic system [19,36,131]. During visual object naming, the brain is engaged in the selection of semantically related names that are competing with each other (names/words represented or belonging to different lexical categories), while during the reading of words, only one phonological information can be uttered [132]. Further, orthographic–phoneme conversion during word reading is faster than semantic conversion in naming processing [19]. Furthermore, different underlying neurocognitive mechanisms might influence naming speed during visual object naming, such as, e.g.., the lexicality effect [57,132], phonological neighborhood effects (dense/sparse objects) [122,123], visual processing [133], and recognition memory [81,134,135]. 

The majority of ERP studies investigating naming processing of adult PDs and children with dyslexia were conducted on non-transparent orthographies using covert lexical decision task with subjects engaged in recognizing visually presented objects, false fonts, strings, visual recognition memory of previously seen words/pseudo words, graphic symbols [81,89,136,137]. These studies found reduced amplitudes of the electric signal in the post-lexical stage (i.e., 500–900 ms) in the left and right hemisphere in adult PDs compared to HCs during visual recognition memory of previously seen/learned words/pseudo words, graphic symbols, and false fonts/consonant strings [81,89,136,137]. Further, regarding ERP studies investigating the naming process of adult PDs and children in the overt design, according to our knowledge, several studies reported findings on non-transparent orthographies in adult PDs [129,133] and children [54]. Araújo, et al. [133] investigated the relationship between how related and unrelated objects were processed on the visual, phonological, and semantic level in adult Portuguese PDs engaged in the sequential covert and overt naming task. The differences were found at a phonological level in a post-lexical stage in the right hemisphere after, i.e., the first 400 ms during overt naming of phonemic related pictures vs. covert naming of phonemic unrelated pictures [133]. Furthermore, Perera et al. [129] reported distributed electrical activity bilaterally over the parieto-occipital region in the left and right hemispheres during overt rapid naming in adult PDs compared to HCs. In the present study, the ERP amplitudes in the post-lexical time window (900–1000 ms) were significantly reduced in the right hemisphere for adult PDs compared to HCs in the overt visual object naming task, similar to the findings of Araujo et al. [133] reporting reduced ERP amplitudes in the right hemisphere in the post-lexical stage during overt naming of phonemic related vs. unrelated pictures in adult PDs compared to HCs. Similar findings of the present study with findings of Araujo et al. [133] could be explained by the previously reported findings of greater ERP amplitudes in the post-lexical stage during covert visual recognition memory [134] of pictures vs. words [135] and overt naming pictures [63] in non-transparent languages in HCs. Therefore, reduced ERP amplitudes in adult PDs compared to HCs during covert visual object naming task could indicate that PDs might use different strategies in recollection processes of previously seen words, pseudo words, objects, and pseudo-objects [81,89,105,134,136], even though behavioral differences in naming response latency were not consistently evident [34,36]. Also, in overt studies, pre-lexical grapho-phonological processes and post-lexical articulatory processes can be assumed to be slightly shifted to later stages in PDs and HCs in transparent and non-transparent orthographies [53,55,56,133]. Therefore, during post-lexical naming process, adult PDs might also probably use the indirect grapho-phonological route to tap phonology and semantics [95].

### 4.3. Study Limitations and Future Directions

There are several limitation factors of the present study, such as the relatively small sample of subjects included in the study (total of 24 subjects) and the small sample of electrodes used for ERP recording. The sample size was determined according to the number of participants included in previously reported ERP studies investigating reading and naming processes in PDs and HCs [62,73,81,83,136,138]. In future studies, according to G*Power 3.1., a sample of at least of 34 subjects in both independent group (dyslexia and control) is recommended to obtain a power of the test of 80%. However, comparable to the present study, several previously reported studies used a similar sample of electrodes, i.e., [57,81,105,121], and included a similar number of PD subjects, i.e., [81,83,138] as in the present study. Further, prior to ERP testing, participants underwent psychological testing and hand dominance tests, while standardized reading tests for adult PDs in Croatian were not available (no available standardized tests in Croatian language for PD subjects). Regarding behavioral measures for inclusion criteria of the HC participants in other ERP studies, few of them used hand dominance and neuropsychological tests similar to the present study [80,99,100]. Furthermore, due to only several previously published studies investigating overt word reading and visual object naming processes in adult PDs in non-transparent languages [56,129,133], make it difficult to compare with the present study’s results. Concerning other methodological issues, the set of words and pictures corresponded (sharing phonological information) in both visual object naming and reading tasks. A similar design has already been used in studies investigating overlapping processes between reading and naming tasks [36,132].

The future ERP studies could further observe cross-cultural electrophysiological correlates of dyslexia in adult PDs in the pre-lexical, lexical, and post-lexical stages of reading and naming processes in other shallow orthographies using overt design and compare these findings with findings for non-transparent orthographies. Also, future studies in different orthographies might observe the multidimensional approach analyzing different neurocognitive processes (rapid automatized naming, word retrieval, visual attention), not solely phonological processing in dyslexia [15,18,84,129,130,131].

## 5. Conclusions

The present ERP study aimed to investigate differences in electrophysiological correlates in the overt reading of words and visual object naming in PDs compared to HCs in the pre-lexical, lexical, and post-lexical processing stages. The results point to the differences in the pre-lexical time window (160–260 ms) in overt reading and the post-lexical time window (900–1000 ms) in overt visual object naming in the right hemisphere in adult PDs compared to HCs. In adult PDs, greater ERP amplitudes were distributed in the right hemisphere in the pre-lexical window in overt, while lowered ERP amplitudes were detected in PDs in the right hemisphere during visual object naming. These study results are supported by findings from two overt studies conducted on non-transparent orthographies reporting greater amplitudes of the electric signal in the right hemisphere in the pre-lexical stage during overt word reading in adult French PDs compared to HCs [56] and reduced amplitudes of the electric signal in post-lexical stage in the right hemisphere during overt naming of phoneme related pictures in adult Portuguese PDs compared to HCs [132].

Overall, the findings of the present study support the view that adult PDs in shallow (transparent orthography) (Croatian language) applying overt design have difficulties: (a) in the pre-lexical stage associated with grapho-phonological processing during reading of words with a combination of sparse neighborhood phonemes, and (b) difficulties in post-lexical stage possible associated with phoneme and word retrieval during overt naming of objects with a combination of sparse neighborhood phonemes.

## Figures and Tables

**Figure 1 bioengineering-11-00459-f001:**
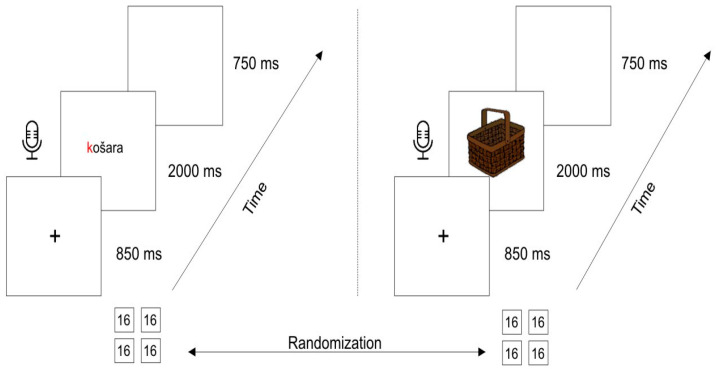
Schematic view of the single trial for reading (**left**) and naming task (**right**). Note: “košara” is the Croatian term for “basket” (English language).

**Figure 2 bioengineering-11-00459-f002:**
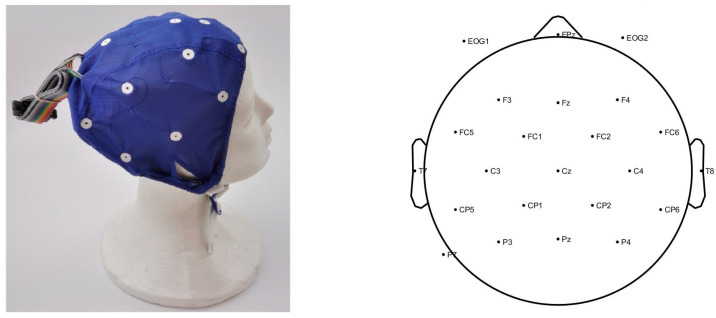
An example of an elastic cap used in the study (**left**), and the channel data location of the electrodes (**right**).

**Figure 3 bioengineering-11-00459-f003:**
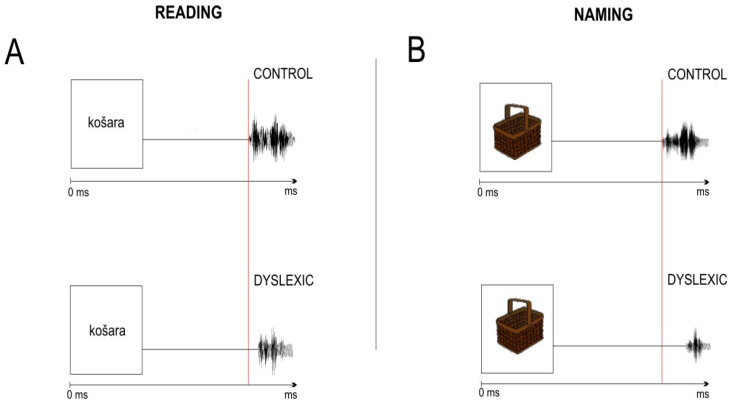
Schematic view of the single trial for word reading (**A**) and visual object naming (**B**) response latency of control (4C) and dyslexia subject (4D). Note: “košara” is the Croatian term for “basket” (English language).

**Figure 4 bioengineering-11-00459-f004:**
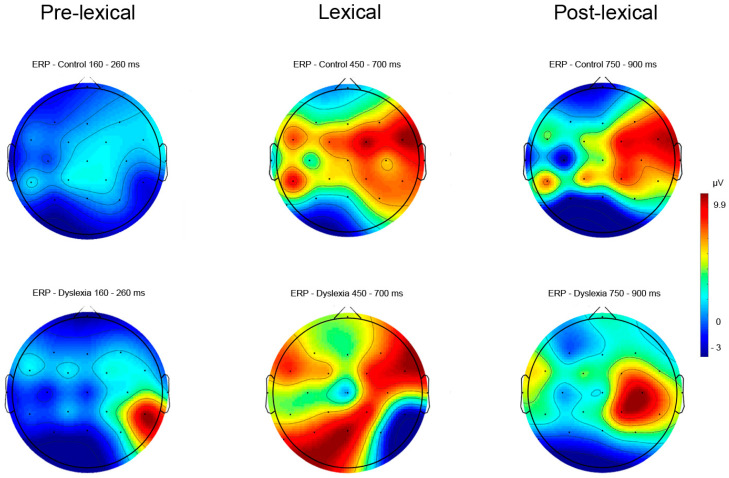
The topographic view of electric signal of word reading ERPs in pre-lexical (160–260 ms), lexical (450–700 ms) and post-lexical (750–900 ms) time windows in PD and HC subjects (color bar represent relative scaling min/max −3/9.9 μV) in control and dyslexic participants.

**Figure 5 bioengineering-11-00459-f005:**
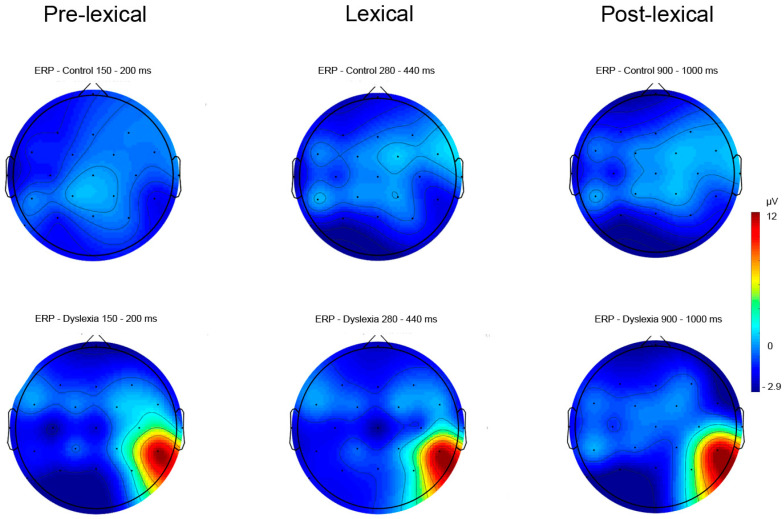
The topographic view of electric signal of object naming ERPs in pre-lexical (150–200 ms), lexical (280–440 ms), and post-lexical (900–1000 ms) time windows in PD and HC subjects (color bar represent relative scaling min/max −2.9/12 μV) in control and dyslexic participants.

**Figure 6 bioengineering-11-00459-f006:**
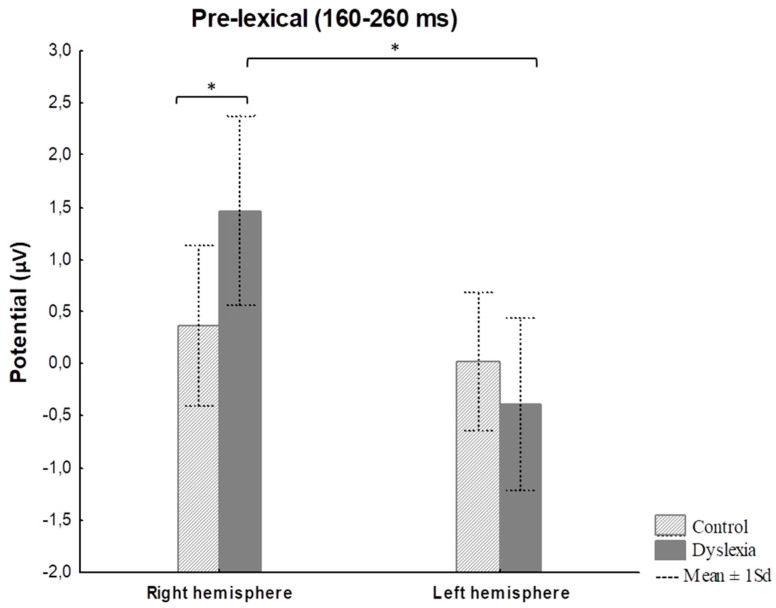
Word reading mean ERP amplitude for a pre-lexical time window (160–260 ms) for the right and left hemispheres in control and dyslexia subjects. * *p* < 0.05. Note: Statistica Soft 12 automatically generates comma sign for decimal separator.

**Figure 7 bioengineering-11-00459-f007:**
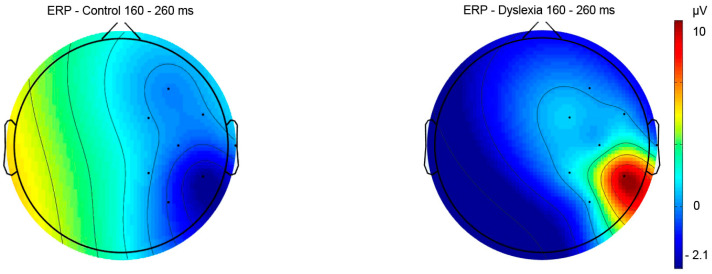
The topographic view of electric signal of right hemisphere electrodes in pre-lexical reading window (160–260 ms) in subjects with dyslexia and control subjects (color bar represents relative scaling min/max –2.1/10 μV) in control and dyslexic participants.

**Figure 8 bioengineering-11-00459-f008:**
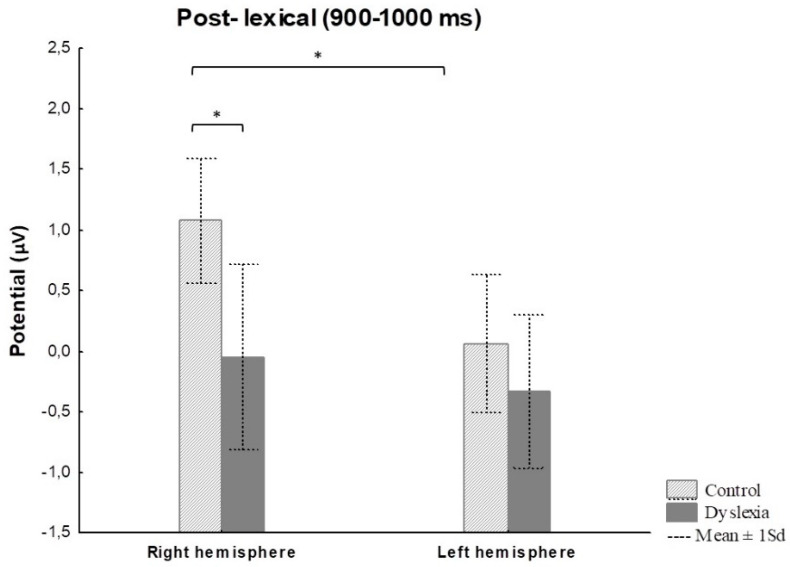
Visual object naming mean ERP amplitude for post-lexical time window (900–1000 ms) for the right and left hemisphere in control and dyslexia subjects. * *p* < 0.05. Note: Statistica Soft 12 automatically generates comma sign for decimal separator.

**Figure 9 bioengineering-11-00459-f009:**
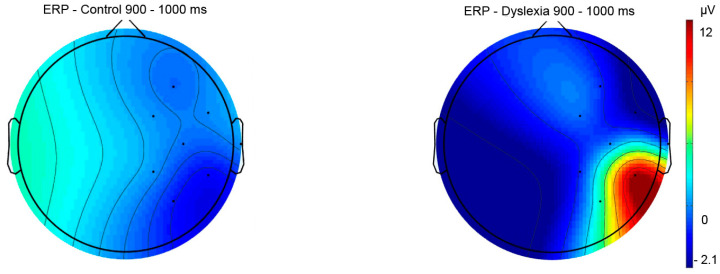
The topographic view of electric signal of the right hemisphere in the post-lexical naming window (900–1000 ms) in control and dyslexia subjects. Color bar represents relative scaling min/max –2.1/12 μV) in control and dyslexic participants.

**Table 1 bioengineering-11-00459-t001:** Qualitative error differences between PD and HC groups for word reading and visual object naming task.

	Qualitative Errors	Control n(%)	Dyslexia n(%)	χ^2^	*p*
Reading	No response	0 (0)	3 (0.39)	3.01	0.08
	Semantic error	0 (0)	0 (0)	-	-
	Phonological error	3 (0.39)	11 (1.43)	4.61	0.03 *
	Hesitation	1 (0.13)	5 (0.65)	2.68	0.10
	Total	4 (0.52)	19 (2.47)	9.93	0.002
Visual object naming	No response	13 (1.72)	77 (10.02)	48.34	<0.001
	Semantic error	125 (16.28)	121 (15.75)	0.08	0.78
	Phonological error	14 (1.82)	26 (3.38)	3.70	0.05
	Hesitation	37 (4.81)	49 (6.38)	1.77	0.18
	Total	189 (24.60)	273 (35.54)	21.84	<0.001

Note: χ^2^—Chi square test; * *p* < 0.05.

**Table 2 bioengineering-11-00459-t002:** Word reading and visual object naming response latency differences for PD and HC subjects.

	Control	Dyslexia	t	*p*
	M ± SD	M ± SD		
Reading latency	663.63 ± 37.09	892.89 ± 96.05	17.81	<0.001
Naming latency	794.44 ± 298.75	803.38 ± 302.03	0.16	0.86

**Table 3 bioengineering-11-00459-t003:** Reading task: two-way ANOVA results of ERPs.

Time Window	Source of Variance	SS	F	*p*	Post hoc Test
Pre-lexical 160–260 ms	Group	8.64	0.54	0.46	HCright-PDright *p* = 0.02 PDright-PDleft *p* < 0.001
	Hemisphere	88.56	13.09	<0.001	
	Group x Hemisphere	41.59	4.79	0.03	
Lexical 450–700 ms	Group	0.23	0.02	0.88	HCright-HCleft *p* = 0.01 PDright-PDleft *p* = 0.01
	Hemisphere	86.32	12.50	<0.001	
	Group x Hemisphere	0.01	0.002	0.96	
Post-lexical 750–900 ms	Group	0.04	0.004	0.94	HCright-HCleft *p* = 0.004
	Hemisphere	57.88	6.82	0.009	
	Group x Hemisphere	15.85	2.07	0.15	

Note: SS—the sum of squares; post hoc test—Fishers LSD test. HC—healthy subjects, PD—people with dyslexia.

**Table 4 bioengineering-11-00459-t004:** Naming task: two-way ANOVA results of ERPs.

Time Window	Source of Variance	SS	F	*p*	Post hoc
Pre-lexical 150–200 ms	Group	8.07	0.32	0.56	HCright-HCleft *p* = 0.04 PDright-PDleft *p* < 0.001
	Hemisphere	220.37	17.03	<0.001	
	Group x Hemisphere	25.27	2.12	0.14	
Lexical 280–440 ms	Group	1.68	0.08	0.77	PDright-PDleft *p* = 0.02
	Hemisphere	77.05	7.94	0.006	
	Group x Hemisphere	0.80	0.08	0.76	
Post-lexical 900–1000 ms	Group	41.77	6.19	0.01	HCright-HCleft *p* = 0.04 HCright-PDright *p* = 0.02
	Hemisphere	30.87	5.62	0.02	
	Group x Hemisphere	9.83	1.44	0.23	

Note: SS—the sum of squares; post hoc test—Fishers LSD test. HC—healthy subjects, PD—people with dyslexia.

## Data Availability

Data are available with a granted proposal upon reasonable request.

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
