# Peer review of "Overt Word Reading and Visual Object Naming in Adults with Dyslexia: Electroencephalography Study in Transparent Orthography"

_bioengineering, 2024, doi:10.3390/bioengineering11050459_

Round 1
Reviewer 1 Report
Comments and Suggestions for Authors
Dear authors,
Concerning the manuscript entitled "Overt Word Reading and Visual Object Naming in Adults with 2 Dyslexia: Electroencephalography Study in Transparent Orthography"
The study is dedicated to investigate overt reading and naming processes in adult people with dyslexia (PDs) in transparent language orthography, by comparing three phases: pre-lexical (150-260 ms), lexi-19 cal (280-700 ms) and post-lexical stage of processing (750-1000 ms) time window. I find your work very interesting and very complete, however it is not mentioned if a sample estimation was conducted beforehand to determine the number of participants in your study. It would be a better study if that was considered before and if more subjects were participated in your study.
The introduction section, the background of your work, I think is complete and well supported with the appropriate references.
In the Materials and Methods section I think that must contain that "The study procedure was approved by the Ethical Committee of the Polyclinic for Rehabilitation of People with Developmental Disorders".
In section 2.3. ERP acquisition and pre-processing you mentioned that "EOG artifacts were removed by automatic eye correction using an ICA algorithm implemented in EEGLAB." My question is ... How effective is this algorithm to trust its outcomes?, then you mentioned that "Furthermore, bad channels were removed directly by an eye inspection or using an algorithm implemented in MATLAB [113]." the same question How effective is this algorithm to trust its outcomes?
Next, in the same section, nowhere is it mentioned if the ERPs were normalized to display the topographic maps in the results section.
In the 3.2. ERP results I have some remarks regarding the visualization of topographic scalpmaps in Figs 4 and 5. and in Figs. 7 and 9, please see the attached files.
Other minor remarks (see the attached file)

Author Response
Dear Reviewer 1,
The answers to the raised concerns are attached in the PDF document.

Reviewer 2 Report
Comments and Suggestions for Authors
“Overt Word Reading and Visual Object Naming in Adults with Dyslexia: Electroencephalography Study in Transparent Orthography”( bioengineering-2958283)
The present investigation aimed to investigate differences in electrophysiological correlates in overt reading and naming in PDs compared to HCs in the pre-lexical, lexical, and post-lexical stages of processing. The results point to the differences in the pre-lexical time window (160-260 ms) in overt reading and the post-lexical time window (900-1000 ms) in overt visual object naming in the right hemisphere in adult PDs compared to HCs. In adult PDs, greater ERP amplitudes were distributed in the right hemisphere in the prelexical window in overt, while lowered ERP amplitudes were observed in PDs in the right hemisphere during visual object naming. Overall, this topic is interesting and the findings hold some practical and theoretical implications. However, some concerns appeared after reading the whole manuscript.
1. Some important papers need to be reviewed and discussed, such as,
Reviews:
Werth, R. Dyslexia: Causes and Concomitant Impairments. Brain Sci. 2023, 13, 472. https://doi.org/10.3390/brainsci13030472
Valdois, S. (2022). The visual‐attention span deficit in developmental dyslexia: Review of evidence for a visual‐attention‐based deficit. Dyslexia, 28(4), 397-415.
Carioti, D., Masia, M. F., Travellini, S., & Berlingeri, M. (2021). Orthographic depth and developmental dyslexia: A meta-analytic study. Annals of Dyslexia, 71(3), 399-438.
Araújo, S., & Faísca, L. (2019). A meta-analytic review of naming-speed deficits in developmental dyslexia. Scientific Studies of Reading, 23(5), 349-368.
Directly related empirical papers:
Premeti, A., Bucci, M. P., Heidlmayr, K., Vigneron, P., & Isel, F. (2024). Neurodynamics of selected language processes involved in word reading: An EEG study with French dyslexic adults. Journal of Neurolinguistics, 71, 101201.
Denis-Noël, A., Colé, P., Bolger, D., & Pattamadilok, C. (2024). How do adults with dyslexia recognize spoken words? Evidence from behavioral and EEG data. Scientific Studies of Reading, 28(1), 21-41.
Silva, P. B., Oliveira, D. G., Cardoso, A. D., Laurence, P. G., Boggio, P. S., & Macedo, E. C. (2022). Event-related potential and lexical decision task in dyslexic adults: Lexical and lateralization effects. Frontiers in Psychology, 13, 852219.
Perera, H., Shiratuddin, M. F., Wong, K. W., & Fullarton, K. (2017, November). EEG signal analysis of passage reading and rapid automatized naming between adults with dyslexia and normal controls. In 2017 8th IEEE International Conference on Software Engineering and Service Science (ICSESS) (pp. 104-108). IEEE.
Araújo, S., Faísca, L., Reis, A., Marques, J. F., & Petersson, K. M. (2016). Visual naming deficits in dyslexia: An ERP investigation of different processing domains. Neuropsychologia, 91, 61-76.
What does the current findings contribute to the Phonological Deficit Theory of dyslexia?
Zoccolotti, P. (2022). Success is not the entire story for a scientific theory: The case of the Phonological Deficit Theory of dyslexia. Brain Sciences, 12(4), 425.
2. “Lately, two studies on dyslexia have been conducted in overt design in adult PDs engaged in reading aloud on words and pseudowords [56] and in children with dyslexia involved in overt rapid automatized naming [54].” What were the findings of these two studies and why the current study is still needed. Moreover, what were the differences between nontransparent languages and languages with shallow (transparent) orthographies and what did you expected the differences might be when it comes to health control and adult PDs? what hypothesis did you want to tested in the current investigation?
3. How did you determine the sample size? Did you calculate the sample size needed before formal study?
The current sample size seems too little to get reliable results.
4. How did you screen adult participants with dyslexia?
5. Please provide a table to depicted the demographic information of both groups and indicate the group differences.
6. As the current study focused on adults with dyslexia, thus the first sentence should not about developmental dyslexia.
7. It is quite strange when you jump the paragraph one to paragraph two. Please pay attention to the flow between paragraphs. Moreover, the first sentence of paragraph two should be revised to better represent the meaning you aimed to express.
8. What does “normal findings” mean in Line 160? Please provide the specific data about this test.
9. Are you sure it is “below 250 Ω” in Line 210?
10. “Hemisphere (Left vs. Right) (Right: F4; FC2; FC6; C4; T8; CP2; CP6; P4; vs. Left: F3; FC5; FC1; T7; C3; CP5; CP1; P7; P3)”about this, did you calculate the average amplitude of F4; FC2; FC6; C4; T8; CP2; CP6 for right hemisphere and the average amplitude of F3; FC5; FC1; T7; C3; CP5; CP1; P7; P3 for left hemisphere? Moreover,why the number of electrodes is different for right (8) and left hemisphere (9)?
11. How did you handle the outliers?
12. Please provide the effect size where available.
13. About the Line 114-130, some repeated information was presented and please make it concise. Moreover, please indicate and summarize what are the specific differences on each ERP component.
14. For the p value, please provide the specific data unless it is less than 0.001.
15. Figure 6 and 8 should be redesigned and the error bars need to be added and the control bar and dyslexia bar should be separated.
16. Please provide the grand-average ERPs at selected ROI electrodes.
17. About the “2.2. Procedure”, please state clearly which window needs the participants response and what will happen after detect the response. How many trials in total for the task.
18. Please provide the materials and stimuli used as supplementary materials. Please provide the descriptive data for ERP as supplementary materials.
19. Please redraw figure 4-6 and keep the color bar the same for both groups under the same condition.
20. In the “4.2. Study limitation” part, the potential future directions need to be stated as well.
21. ‘As far as we know, the present ERP study is the first one exploring the reading and naming process in overt design in shallow orthography in adult PDs.’ The novelty statement needs to be moved to the introduction part.
22. I recommend that the paper be thoroughly proofread and edited for languages and grammars, to enhance readership.
Comments on the Quality of English Language
Moderate editing of English language required
Author Response
Dear Reviewer 2,
The answers to the raised concerns are attached in the PDF document.

Reviewer 3 Report
Comments and Suggestions for Authors
Report on the manuscript “Overt Word Reading and Visual Object Naming in Adults with Dyslexia: Electroencephalography Study in Transparent Orthography”
I rate this work quite highly and recommend it for publication after minor revisions.
1.
Line 18:
“The results of adult (PDs)…”
It is better to remove the parentheses:
“The results of adult PDs…”
2.
Line 22:
“ERP amplitude for prelexical, lexical, and post-lexical time window, a mixed design ANOVA was…”
Please decode the abbreviation “ERP” here in the abstract or keywords.
Please add a brief explanation for readers of what ANOVA is somewhere in the main text.
3.
Lines 74-75 and 76:
“The dual route cascaded model of visual word recognition and reading aloud (DCR) [19]…”
“According to a mostly accepted DCR model [19]…”
Please use the same abbreviation as in [19] and change “DCR” to “DRC”.
I hope this was a hidden dyslexia test for the reviewers.
4.
Lines 193-194:
“The entire experimental session lasted approximately 45 minutes for each subject.”
Lines 201-202:
“Participants were asked to avoid eye blinks and to be calm during recording.”
How long should participants not blink?
5.
Line 277 and similar:
“differences in word reading latency (t=17.81; p<0.001) between PD and HC sub- 277><0.001) between PD and HC”
What are “t” and “p”? What are their units?
6.
Figures 4 and 5. “The topographic view of …”
Are these images of a representative object or averaged color maps?
7.
Lines 315-316 and similar:
“left hemisphere for prelexical (Fhemisphere = 13.09, p < 0.001; Fgroup x hemisphere = 4.79, p < 0.05)”
What are “Fhemisphere” and “Fgroup x hemisphere”? What are their units?
8.
Table 3, second row, last column, line 336
“HCright-PDright p<0.05”
Is this really “HCright-PDright”? Or “HCright-HDright”?
Please answer my questions in the main text, not just in a reply to the reviewer.
Last, but perhaps the most important note.
The authors describe the difference in brain activation between healthy people and people with dyslexia as an observed fact. I would be glad if the authors could add some discussion about the mechanism of this difference.
Author Response
Dear Reviewer 3,
The answers to the raised concerns are attached in the PDF document.

Round 2
Reviewer 2 Report
Comments and Suggestions for Authors
Thanks for the revisions and some concerns remain.
1. About the sample size determination, if you did not calculate the required sample size before formal experiment, then you need to state clearly how did you determine the sample size in the current investigation, you mentioned in the response letter that you referred to the previous related studies, it is also acceptable. Moreover, the current sample size did not meet the minimum required sample size as calculated by gpower, which need to be included as a potential limitation and larger sample size needs to be encouraged to be included in the future to replicate the current findings.
2. For the revised figure 6 and 8, please indicate the ±1 SD for the dyslexia group as control group did and please state the meaning of the error bar in the figure caption.
About the outliers, I also mean the behavioral performance, how did you treat them?
3. “4.2. Study limitation” can be revised to “4.2. Study limitations and future directions” and include the part about future directions in the conclusion part. The conclusion part needs to be more concise and about one paragraph to summarize the main findings and implications of current results would be enough.
4. About the effect size, please provide the d value for the t test and the η2 for the ANOVA results.
5. For the p value, please provide the specific data unless it is less than 0.001. Please check all the context and all the tables.
Comments on the Quality of English LanguageMinor editing of English language required
Author Response
We want to express our gratitude to the honourable Reviewer for further revision of our manuscript. In the author’s reply to concerns, we have responded to the Reviewer’s comments and suggestions and updated the manuscript accordingly with changes highlighted in yellow.
